# ShiftAddNet: A Hardware-Inspired Deep Network

**Haoran You**[†], **Xiaohan Chen**[‡], **Yongan Zhang**[†], **Chaojian Li**[†], **Sicheng Li**[◇], **Zihao Liu**[◇],
**Zhangyang Wang**[‡], and **Yingyan Lin**[†]

[†]Department of Electrical and Computer Engineering, Rice University
[‡]Department of Electrical and Computer Engineering, The University of Texas at Austin
[◇]Alibaba DAMO Academy
[†]*{hy34, yz87, cl114, yingyan.lin}@rice.edu*, [‡]*{xiaohan.chen, atlaswang}@utexas.edu*
[◇]*{sicheng.li, zihao.liu}@alibaba-inc.com*

## Abstract

Multiplication (e.g., convolution) is arguably a cornerstone of modern deep neural networks (DNNs). However, intensive multiplications cause expensive resource costs that challenge DNNs' deployment on resource-constrained edge devices, driving several attempts for multiplication-less deep networks. This paper presented **ShiftAddNet**, whose main inspiration is drawn from a common practice in energy-efficient hardware implementation, that is, multiplication can be instead performed with additions and logical bit-shifts. We leverage this idea to explicitly parameterize deep networks in this way, yielding a new type of deep network that involves only bit-shift and additive weight layers. This *hardware-inspired* ShiftAddNet immediately leads to both energy-efficient inference and training, without compromising the expressive capacity compared to standard DNNs. The two complementary operation types (bit-shift and add) additionally enable finer-grained control of the model's learning capacity, leading to more flexible trade-off between accuracy and (training) efficiency, as well as improved robustness to quantization and pruning. We conduct extensive experiments and ablation studies, all backed up by our FPGA-based ShiftAddNet implementation and energy measurements. Compared to existing DNNs or other multiplication-less models, ShiftAddNet aggressively reduces **over 80%** hardware-quantified energy cost of DNNs training and inference, while offering comparable or better accuracies. Codes and pre-trained models are available at https://github.com/RICE-EIC/ShiftAddNet.

## 1 Introduction

Powerful deep neural networks (DNNs) come at the price of prohibitive resource costs during both DNN inference and training, limiting the application feasibility and scope of DNNs in resource-constrained device for more pervasive intelligence. DNNs are largely composed of multiplication operations for both forward and backward propagation, which are much more computationally costly than addition [1]. The above roadblock has driven several attempts to design new types of *hardware-friendly* deep networks, which rely less on heavy multiplications in order for higher energy efficiency. ShiftNet [2, 3] adopted spatial shift operations paired with pointwise convolutions, to replace a large portion of convolutions. DeepShift [4] employed an alternative of bit-wise shifts, which are equivalent to multiplying the input with powers of 2. Lately, AdderNet [5] pioneered to demonstrate the feasibility and promise of replacing *all convolutions* with merely addition operations.

This paper takes one step further along this direction of multiplication-less deep networks, by drawing a very fundamental idea in the hardware-design practice, computer processors, and even digital signal processing. It has been known for long that multiplications can be performed with additions and logical bit-shifts [6, 7], whose hardware implementation are very simple and much faster [8], without compromising the result quality or precision. Also on currently available processors, a bit-shift instruction is faster than a multiply instruction and can be leveraged to multiply (shift left) and divide

(shift right) by powers of two. Multiplication (or division) by a constant is then implemented using a sequence of shifts and adds (or subtracts). The above clever "shortcut" saves arithmetic operations, and can readily be applied to accelerating the hardware implementation of any machine learning algorithm involving multiplication (either scalar, vector, or matrix). But our curiosity is well beyond this: *can we learn from this hardware-level "shortcut" to design efficient learning algorithms?*

The above uniquely motivates our work: in order to be more "hardware-friendly", we strive to re-design our model to be "**hardware-inspired**", leveraging the successful experience directly form the efficient hardware design community. Specifically, we explicit re-parameterize our deep networks by replacing all convolutional and fully-connected layers (both built on multiplications) with two multiplication-free layers: bit-shift and add. Our new type of deep model, named **ShiftAddNet**, immediately lead to both energy-efficient inference and training algorithms.

We note that ShiftAddNet seamlessly integrates bit-shift and addition together, with strong motivations that address several pitfalls in prior arts [2, 3, 4, 5]. Compared to utilizing spatial- or bit-shifts alone [2, 4], ShiftAddNet can be fully expressive as standard DNNs, while [2, 4] only approximate the original expressive capacity since shift operations cannot span the entire continuous space of multiplicative mappings (e.g., bit-shifts can only represent the subset of power-of-2 multiplications). Compared to the fully additive model [5], we note that while repeated additions can in principle replace any multiplicative mapping, they do so in a very inefficient way. In contrast, by also exploiting bit-shifts, ShiftAddNet is expected to be more parameter-efficient than [5] which relies on adding templates. As a bonus, we notice the bit-shift and add operations naturally correspond to coarse- and fine-grained input manipulations. We can exploit this property to more flexibly trade-offs between training efficiency and achievable accuracy, e.g., by freezing all bit-shifts and only training add layers. ShiftAddNet with fixed shift layers can achieve up to 90% and 82.8% energy savings than fully additive models [5] and shift models [4] under floating-point or fixed-point precisions trainings, while leading to a comparable or better accuracies (-3.7% $\sim$ +31.2% and 3.5% $\sim$ 23.6%), respectively. Our contributions can be summarized as follow:

- Uniquely motivated by the hardware design expertise, we combine two multiplication-less and complementary operations (bit-shift and add) to develop a hardware-inspired network called ShiftAddNet that is fully expressive and ultra-efficient.

- We develop training and inference algorithms for ShiftAddNet. Leveraging the two operations' distinct granularity levels, we also investigate ShiftAddNet trade-offs between training efficiency and achievable accuracy, e.g., by freezing all the bit-shift layers.

- We conduct extensive experiments to compare ShiftAddNet with existing DNNs or multiplication-less models. Results on multiple benchmarks demonstrate its superior compactness, accuracy, training efficiency, and robustness. Specifically, we implement ShiftAddNet on a ZYNQ-7 ZC706 FPGA board [9] and collect all real energy measurements for benchmarking.

## 2 Related works

**Multiplication-less DNNs.** Shrinking the cost-dominate multiplications has been widely considered in many DNN designs for reducing the computational complexity [10, 11]: [10] decomposes the convolutions into separate depthwise and pointwise modules which require fewer multiplications; and [12, 13, 14] binarize the weights or activations to construct DNNs consisting of sign changes paired with much fewer multiplications. Another trend is to replace the multiplication operations with other cheaper operations. Specifically, [3, 2] leverage spatial shift operations to shift feature maps, which needs to be cooperated with pointwise convolution to aggregate spatial information; [4] fully replaces multiplications with both bit-wise shift operations and sign changes; and [5, 15, 16] trade multiplications for cheaper additions and develop a special backpropogation scheme for effectively training the add-only networks.

**Hardware costs of basic operations.** As compared to shift and add, multipliers can be very inefficient in hardware as they require high hardware costs in terms of consumed energy/time and chip area. Shift and add operations can be a substitute for such multipliers. For example, they have been adopted for saving computer resources and can be easily and efficiently performed by a digital processor [17]. This hardware idea has been adopted to accelerate multilayer perceptrons (MLP) in digital processors [8]. We here motivated by such hardware expertise to fully replace multiplications in modern DNNs with merely shift and add, aiming to solve the drawbacks in existing shift-only or add-only replacements methods and to boost the network efficiency over multiplication-based DNNs.

**Relevant observations in DNN training.** It has been shown that DNN training contains redundancy in various aspects [18, 19, 20, 21, 22, 23]. For example, [24] explores an orthogonal weight training algorithm which over-parameterizes the networks with the multiplication between a learnable orthogonal matrix and fixed randomly initialized weights, and argue that fixing weights during training and only learning a proper coordinate system can yield good generalization for over-parameterized networks; and [25] separates the convolution into spatial and pointwise convolutions, while freezing the binary spatial convolution filters (called anchor weights) and only learning the pointwise convolutions. These works inspire the ablation study of fixing shift parameters in our ShiftAddNet.

## 3 The proposed model: ShiftAddNet

In this section, we present our proposed ShiftAddNet. First, we will introduce the motivation and hypothesis beyond ShiftAddNet, and then discuss ShiftAddNet's component layers (i.e., shift and add layers) from both hardware cost and algorithmic perspectives, providing high-level background and justification for ShiftAddNet. Finally, we discuss a more efficient variant of ShiftAddNet.

### 3.1 Motivation and hypothesis

Driven from the long-standing tradition in the field of energy-efficient hardware implementation to replace expensive multiplication with lower-cost bit-shifts and adds, we re-design DNNs by pipelining the shift and add layers. We hypothesize that (1) while DNNs with merely either shift and add layers in general are less capable compared to their multiplication-based DNN counterparts, integrating these two weak players can lead to networks with much improved expressive capacity, while maintaining their hardware efficient advantages; and (2) thanks to the coarse- and fine-grained input manipulations resulted from the complementary shift and add layers, there is a possibility that such new network pipeline can even lead to new models which are comparable with multiplication-based DNNs in terms of task accuracy, while offering superior hardware efficiency.

### 3.2 ShiftAddNet: shift Layers

This subsection discusses the shift layers adopted in our proposed ShiftAddNet in terms of hardware efficiency and algorithmic capacity.

**Hardware perspective.** Shift operation is a well known efficient hardware primitive, motivating recent development of various shift-based efficient DNNs [4, 2, 3]. Specifically, the shift layers in [4] reduce DNNs' computation and energy costs by replacing the regular cost-dominant multiplication-based convolution and linear operations (a.k.a fully-connected layers) with bit-shift-based convolution and linear operations, respectively. Mathematically, such bit-

Table 1: Unit energy comparisons using ASIC & FPGA.

| Format | | ASIC (45nm) | | FPGA (ZYNQ-7 ZC706) | |
|---|---|---|---|---|---|
| Operation | Format | Energy (pJ) | Improv. | Energy (pJ) | Improv. |
| Mult. | FP32 | 3.7 | - | 18.8 | - |
| | FIX32 | 3.1 | - | 19.6 | - |
| | FIX8 | 0.2 | - | 0.2 | - |
| Add | FP32 | 0.9 | **4.1x** | 0.4 | **47x** |
| | FIX32 | 0.1 | **31x** | 0.1 | **196x** |
| | FIX8 | 0.03 | **6.7x** | 0.1 | **2x** |
| Shift | FIX32 | 0.13 | **24x** | 0.1 | **196x** |
| | FIX8 | 0.024 | **8.3x** | 0.025 | **8x** |

shift operations are equivalent to multiplying by powers of 2. As summarized in Tab. 1, such shift operations can be extremely efficient as compared to their corresponding multiplications. In particular, bit-shifts can save as high as $196\times$ and $24\times$ energy costs over their multiplication couterpart, when implemented in a 45nm CMOS technology and SOTA FPGA [26], respectively. In addition, for a 16-bit design, it has been estimated that the average power and area of multipliers are at least $9.7\times$ and $1.45\times$, respectively, of the bit-shifts [4].

**Algorithmic perspective.** Despite its promising hardware efficiency, networks constructed with bit-shifts can compare unfavorably with its multiplication-based counterpart in terms of expressive efficiency. Formally, expressive efficiency of architecture A is higher than architecture B if any functions realized by B could be replicated by A, but there exists functions realized by A, which cannot be replicated by B unless its size grows significantly larger [27]. For example, it is commonly adopted that DNNs are exponentially efficient as compared to shallow networks because a shallow network must grow exponentially large for approximating the functions represented by a DNN of polynomial sizes. For ease of discussion, we refer to [28] and use a loosely defined metric of expressiveness called expressive capacity (accuracy) in this paper without loss of generality. Specifically, expressive capacity refers to the achieved accuracy of networks under the same or similar hardware cost, i.e., network A is deemed to have a better expressive capacity compared to network B if the former achieves a higher accuracy at a cost of the same or even fewer FLOPs (or energy cost). From this

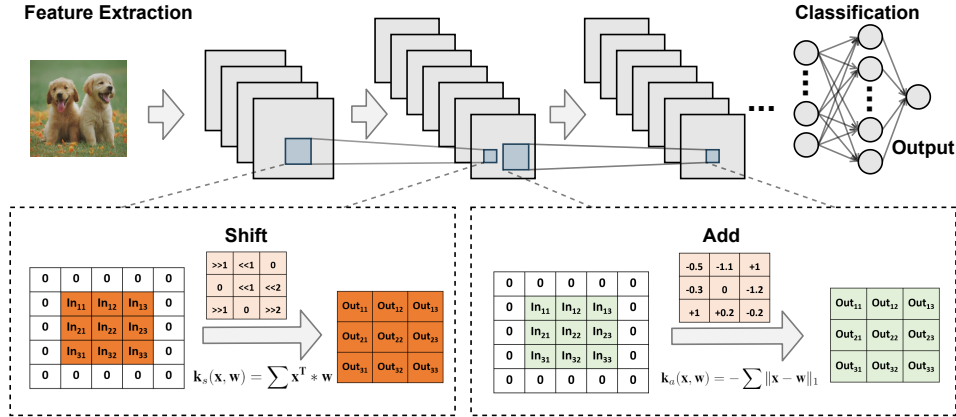

Figure 1: Illustrating the overview structure of ShiftAddNet.

perspective, networks with bit-shift layers without full-precision latent weights are observed to be inferior to networks with add layers or multiplication-based convolution layers as shown in prior arts [5, 10] and validated in our experiments (see Sec. 4) under various settings and datasets.

### 3.3 ShiftAddNet: add Layers

Similar to the aforementioned subsection, here we discuss the add layers adopted in our proposed ShiftAddNet in terms of hardware efficiency and algorithmic capacity.

**Hardware perspective.** Addition is another well known efficient hardware primitive. This has motivated the design of efficient DNNs mostly using additions [5], and there are many works trying to trade multiplications for additions in order to speed up DNNs [29, 5, 30]. In particular, [5] investigates the feasibility of replacing multiplications with additions in DNNs, and presents AdderNets which trade the massive multiplications in DNNs for much cheaper additions to reduce computational costs. As a concrete example in Tab. 1, additions can save up to $196\times$ and $31\times$ energy costs over multiplications in fixed-point formats, and can save $47\times$ and $4.1\times$ energy costs in flaoting-point formats (more expensive), when being implemented in a 45nm CMOS technology and SOTA FPGA [26], respectively. Note that the pioneering work [5] which investigates addition dominant DNNs presents their networks in flaoting-point formats.

**Algorithmic perspective.** While there have been no prior works studying add layers in terms of expressive efficiency or capacity. The results in SOTA bit-shift-based networks [4] and add-based networks [5] as well as our experiments under various settings show that add-based networks in general have better expressive capacity than their bit-shift-based counterparts. In particular, AdderNets [5] achieves a 1.37% higher accuracy than that of DeepShift [4] at a cost of similar or even lower FLOPs on ResNet-18 with the ImageNet dataset. Furthermore, when it comes to DNNs, the diversity of learned features' granularity is another factor that is important to the achieved accuracy [31]. In this regard, shift layers are deemed to extract large-grained feature extraction as compared to small-grained features learned by add layers.

### 3.4 ShiftAddNet implementation

#### 3.4.1 Overview of the structure

To better validate our aforementioned Hypothesis (1), i.e., integrating the two weak players (shift and add) into one can lead to networks with much improved task accuracy and hardware efficiency as compared to networks with merely one of the two weak players, we adopt SOTA bit-shift-based and add-based networks' design to implement the shift and add layers of our ShiftAddNet in this paper. In this way, we can better evaluate that the resulting designs' improved performance comes merely from the integration effect of the two, ruling out potential impact due to a different design. The overall structure of ShiftAddNet is illustrated in Fig. 1, which can be formulated as follows:

$$\boldsymbol{O} = \mathbf{k}_a(\mathbf{k}_s(\boldsymbol{I}, \mathbf{s} \cdot 2^{\mathbf{p}}), \mathbf{w}_a), \quad \text{where } \mathbf{k}_s(\mathbf{x}, \mathbf{w}) = \sum \mathbf{x}^{\mathbf{T}} * \mathbf{w} \text{ and } \mathbf{k}_a(\mathbf{x}, \mathbf{w}) = -\sum \|\mathbf{x} - \mathbf{w}\|_1 \quad (1)$$

where $\boldsymbol{I}$ and $\boldsymbol{O}$ denote the input and output activations, respectively; $\mathbf{k}_s(\cdot, \cdot)$ and $\mathbf{k}_a(\cdot, \cdot)$ are kernel functions to perform the inner products and subtractions of a convolution; $\mathbf{w}_a$ denotes the weights

in the add layers, and $\mathbf{w}_s = \mathbf{s} \cdot 2^{\mathbf{p}}$ represents the weights in the shift layers, in which $\mathbf{s}$ are sign flip operators $\mathbf{s} \in \{-1, 0, 1\}$ and the powers of 2 parameters $\mathbf{p}$ can represent the bit-wise shift.

**Dimensions of shift and add layers.** A shift layer in ShiftAddNet adopts the same strides and weight dimensions as that in the corresponding multification-based DNNs (e.g., ConvNet), followed by an add layer which adapts its kernel sizes and input channels to match the reduced feature maps. Although in this way ShiftAddNet contains slightly more weights than that of ConvNet/AdderNet (e.g., $\sim$1.3MB vs. 1.03 MB in ConvNet/AdderNet (FP32) on ResNet20), it consumes less energy costs to achieve similar accuracies because data movement is the cost bottleneck in hardware acceleration [32, 33, 34]. ShiftAddNet can be further quantized to 0.4 MB (FIX8) without hurting the accuracy, which will be demonstrated using experiments in Sec. 4.2.

### 3.4.2 Backpropagation in ShiftAddNet

ShiftAddNet adopts SOTA bit-shift-based and add-based networks' design during backpropagation. Here we explicitly formulate both the inference and backpropagation of the shift and add layers. The add layers during inference can be expressed as:

$$
\begin{aligned}
\boldsymbol{O}_a[c_o][e][f] &= -\sum \|\mathbf{x}_a - \mathbf{w}_a\|_1 \\
&= -\sum_{c_i=0}^{C_I-1} \sum_{r=0}^{R-1} \sum_{s=0}^{S-1} \|\mathbf{x}_a[c_i][e+r][f+s] - \mathbf{w}_a[c_o][c_i][r][s]\|_1,
\end{aligned}
\tag{2}
$$

where $0 \le c_o < C_O$, $0 \le e < E$, $0 \le f < F$, and specifically, $C_I$ and $C_O$, $E$ and $F$, $R$ and $S$ stand for the number of the input and output channels, the size of the input and output feature maps, and the size of the weight filters, respectively; and $\boldsymbol{O}_a$, $\mathbf{x}_a$, and $\mathbf{w}_a$ denote the output and input activations, and the weights, respectively. Based on the above notation, we formulate the add layers' backpropagation in the following equations:

$$
\frac{\partial \boldsymbol{O}_a[c_o][e][f]}{\partial \mathbf{w}_a[c_o][c_i][r][s]} = \mathbf{x}_a[c_i][e+r][f+s] - \mathbf{w}_a[c_o][c_i][r][s],
\tag{3}
$$

$$
\frac{\partial \boldsymbol{O}_a[c_o][e][f]}{\partial \mathbf{x}_a[c_i][e+r][f+s]} = \mathrm{HT}(\mathbf{x}_a[c_i][e+r][f+s] - \mathbf{w}_a[c_o][c_i][r][s]),
\tag{4}
$$

where HT denotes the HardTanh function following AdderNet [5] to prevent gradients from exploding. Note that the difference over AdderNet is that the strides of ShiftAddNet's add layers are always equal to one, while its shift layers share the same strides with its corresponding ConvNet.

Next, we use the above notation to introduce both the inference and backpropagation design of ShiftAddNet's shift layers, with one additional symbol $U$ denoting the stride:

$$
\boldsymbol{O}_s[c_o][e][f] = \sum \mathbf{x}_s^{\mathbf{T}} * \mathbf{w}_s = \sum_{c_i=0}^{C_I-1} \sum_{r=0}^{R-1} \sum_{s=0}^{S-1} \mathbf{x}_s[c_i][eU+r][fU+s] \cdot \mathbf{s}[c_o][c_i][r][s] \cdot 2^{\mathbf{p}[c_o][c_i][r][s]},
\tag{5}
$$

$$
\frac{\partial \boldsymbol{O}_a}{\partial \mathbf{p}} = \frac{\partial \boldsymbol{O}_a}{\partial \boldsymbol{O}_s} \frac{\partial \boldsymbol{O}_s}{\partial \mathbf{w}_s} \cdot \mathbf{w}_s \cdot \ln 2, \quad \frac{\partial \boldsymbol{O}_a}{\partial \mathbf{s}} = \frac{\partial \boldsymbol{O}_a}{\partial \boldsymbol{O}_s} \frac{\partial \boldsymbol{O}_s}{\partial \mathbf{w}_s}, \quad \frac{\partial \boldsymbol{O}_a}{\partial \mathbf{x}_s} = \frac{\partial \boldsymbol{O}_a}{\partial \boldsymbol{O}_s} \cdot \mathbf{w}_s^{\mathbf{T}},
\tag{6}
$$

where $\boldsymbol{O}_s$, $\mathbf{x}_s$, and $\mathbf{w}_s$ denote the output and input activations, and the weights of shift layers, respectively; $\partial \boldsymbol{o}_a / \partial \boldsymbol{o}_s$ follows Equ. (4) to perform backpropagation since a shift layer is followed by an add layer in ShiftAddNet.

### 3.4.3 ShiftAddNet variant: fixing the shift layers

Inspired by the recent success of freezing anchor weights [25, 24] for over-parameterized networks, we hypothesize that freezing the "over-parameterized" shift layers (large-grained anchor weights) in ShiftAddNet can potentially lead to a good generalization ability, motivating us to develop a variant of ShiftAddNet with fixed shift layers. In particular, ShiftAddNet with fixed shift layers simply means the shift weight filters $\mathbf{s}$ and $\mathbf{p}$ in Equ. (1) remain the same after initialization. Training ShiftAddNet with fixed shift layers is straightforward because the shift weight filters (i.e., $\mathbf{s}$ and $\mathbf{p}$ in Equ.(1)) do not need to be updated (i.e., skipping the corresponding gradient calculations) while the error can be backpropagated through the fixed shift layers in the same way as they are backpropagated through the learnable shift layers (see Equ. (6)). Moreover, we further prune the fixed shift layers to only reserve the necessary large-grained anchor weights to design a more energy-efficient ShiftAddNet.

# 4 Experiment results

In this section, we first describe our experiment setup, and then benchmark ShiftAddNet over SOTA DNNs. After that, we evaluate ShiftAddNet in the context of domain adaptation. Finally, we present ablation studies of ShiftAddNet's shift and add layers.

## 4.1 Experiment setup

**Models and datasets.** We consider two DNN models (i.e., ResNet-20 [35] and VGG19-small models [36]) on six datasets: two classification datasets (i.e., CIFAR-10/100) and four IoT datasets (including MHEALTH [37], FlatCam Face [38], USCHAD [39], and Head-pose detection [40]). Specifically, the Head-pose dataset contains 2,760 images, and we adopt randomly sampled 80% for training and the remaining 20% for testing the correctness of three outputs: front, left, and right [41]; the FlatCam Face dataset contains 23,838 face images captured using a FlatCam lensless imaging system [38], which are resized to $76 \times 76$ before being used.

**Training settings.** For the CIFAR-10/100 and Head-pose datasets, the training takes a total of 160 epochs with a batch size of 256, where the initial learning rate is set to 0.1 and then divided by 10 at the 80-th and 120-th epochs, respectively, and a SGD solver is adopted with a momentum of 0.9 and a weight decay of $10^{-4}$ following [42]. For the FlatCam Face dataset, we follow [43] to pre-train the network on the VGGFace 2 dataset for 20 epochs before adapting to the FlatCam Face images. For the trainable shift layers, we follow [4] to adopt 50% sparsity by default; and for the MHEALTH and USCHAD datasets, we follow [44] to use a DCNN model and train it for 40 epochs.

**Baselines and evaluation metrics.** Baselines: We evaluate the proposed ShiftAddNet over two SOTA multiplication-less networks, including AdderNet [5] and DeepShift (use DeepShift (PS) by default) [4], and also compare it to the multiplication-based ConvNet [45] under a comparable energy cost ($\pm$ 30% more than AdderNet (FP32)). Evaluation metrics: For evaluating real hardware efficiency, we measure the energy cost of all DNNs on a SOTA FPGA platform, ZYNQ-7 ZC706 [9]. Note that our energy measurements in all experiments include the DRAM access costs.

## 4.2 ShiftAddNet over SOTA DNNs on standard training

**Experiment settings.** For this set of experiments, we consider the general ShiftAddNet with learnable shift layers. For the two SOTA multiplication-less baselines: AdderNet [5] and DeepShift [4], the latter of which quantizes its activations to 16-bit fixed-point for shifting purposes while its back-propagation uses a floating-point precision. As floating-point additions are more expensive than multiplications [46], we refer to SOTA quantization techniques [47, 48] for quantizing both the forward (weights and activations) and backward (errors and gradients) parameters to 32/8-bit fixed-point (FIX32/8), for evaluating the potential energy savings of both the ShiftAddNet and AdderNet.

**ShiftAddNet over SOTA on classification.** The results on four datasets and two DNNs in Fig. 2 (a), (b), (e), and (d) show that ShiftAddNet can **consistently outperform all competitors** in terms of the measured energy cost, while improving the task accuracies. Specifically, with full-precision floating-point (FP32) ShiftAddNet even surpasses both the multiplication-based ConvNet and the AdderNet: when training ResNet-20 on CIFAR-10, ShiftAddNet reduces 33.7% and 44.6% of the training energy costs as compared to AdderNet and ConvNet [45], respectively, outperforming SOTA multiplication-based ConvNet and thus validating our Hypothesis (2) in Section 3.1; and ShiftAddNet demonstrates **notably improved robustness to quantization as compared to AdderNet**: a quantized ShiftAddNet with 8-bit fixd-point presentation reduces 65.1% $\sim$ 75.0% of the energy costs over the reported one of AdderNet (with a floating-point precision, as denoted as FP32) while offering comparable accuracies (-1.79% $\sim$ +0.18%), and achieves a greatly higher accuracy (7.2% $\sim$ 37.1%) over the quantized AdderNet (FIX32/8) while consuming comparable or even less energy costs (-25.2% $\sim$ 25.2%). Meanwhile, ShiftAddNet achieves 2.41% $\sim$ 16.1% higher accuracies while requiring 34.1% $\sim$ 70.9% less energy costs, as compared to DeepShift [4]. This set of results also verify our Hypothesis (1) in Section 3.1 that integrating the weak shift and add players can lead to improved network expressive capacity with negligible or even lower hardware costs.

We also compare ShiftAddNet with the baselines in an apple-to-apple manner based on the same quantization format (e.g., FIX32). For example, when evaluated on VGG-19 with CIFAR-10 (see Fig. 2 (c)), ShiftAddNet consistently (1) improves the accuracuies by 11.6%, 10.6%, and 37.1% as compared to AdderNet in FIX32/16/8 formats, at comparable energy costs (-25.2% $\sim$ 15.7%); and (2) improves the accuracies by 26.8%, 26.2%, and 24.2% as compared to DeepShift (PS) using FIX32/16/8 formats, with comparable or slightly higher energy overheads. To further

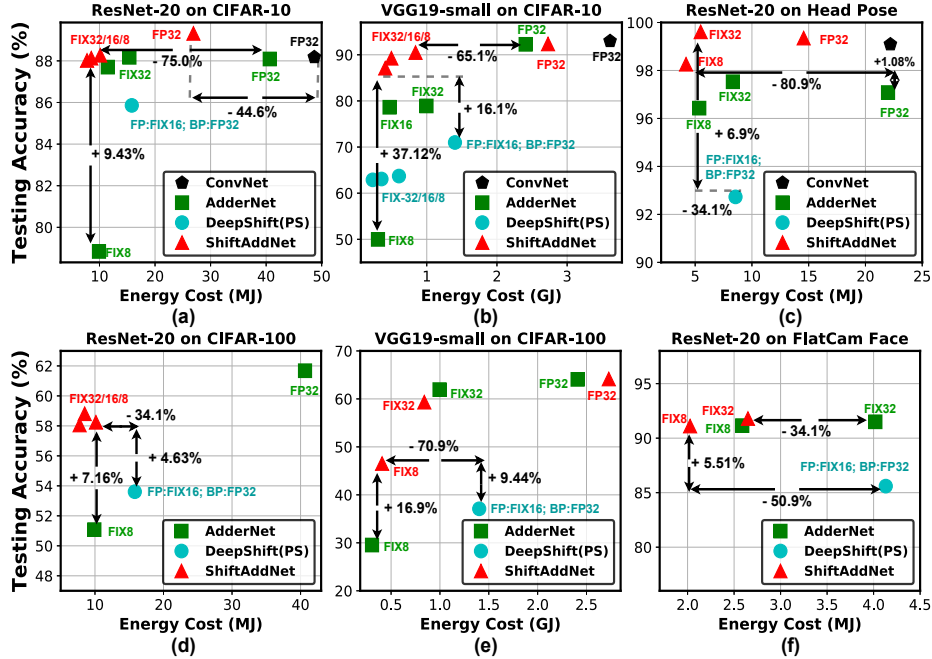

Figure 2: Tesing accuracy vs. energy cost of ShiftAddNet over AdderNet [5] (add only), DeepShift [4] (shift only), and multiplication-based ConvNet [45], using ResNet-20 and VGG19-small models on CIFAR-10/100 and two IoT datasets.

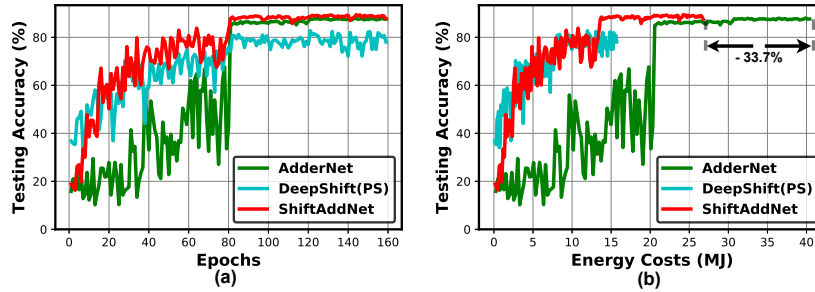

Figure 3: Testing accuracy's trajectories visualization for ShiftAddNet, AdderNet [5], and DeepShift [4] versus both training epochs and energy costs when evaluated on ResNet-20 with CIFAR-10.

analyze ShiftAddNet's improved robustness to quantization, we compare the discriminative power of AdderNet and ShiftAddNet by visualizing the class divergences using the $t$-SNE algorithm [49], as shown in the supplement.

**ShiftAddNet over SOTA on IoT applications.** We further evaluate ShiftAddNet over the SOTA baselines on the two IoT datasets to evaluate its effectiveness on real-world IoT tasks. As shown in Fig. 2 (c) and (f), ShiftAddNet again consistently outperforms the baselines under all settings in terms of efficiency-accuracy trade-offs. Specifically, compared with AdderNet, ShiftAddNet achieves 34.1% $\sim 80.9\%$ energy cost reductions while offering 1.08% $\sim 3.18\%$ higher accuracies; and compared with DeepShift (PS), ShiftAddNet achieves 34.1% $\sim 50.9\%$ energy savings while improving accuracies by 5.5% $\sim 6.9\%$. This set of experiments show that ShiftAddNet's effectiveness and superiority extends to read-world IoT applications. We also observe similar improved efficiency-accuracy trade-offs on the MHEALTH [37] and USCHAD [39] datasets and report the performance in the supplement.

**ShiftAddNet over SOTA on training trajectories.** Fig. 3 (a) and (b) visualize the testing accuracy's trajectories of ShiftAddNet and the two baselines versus both the training epoch and energy cost, respectively, on ResNet-20 with CIFAR-10. We can see that ShiftAddNet achieves a comparable or higher accuracy with fewer epochs and energy costs, indicating its better generalization capability.

## 4.3 ShiftAddNet over SOTA on domain adaption and fine-tuning

To further evaluate the potential capability of ShiftAddNet for on-device learning [50], we consider the training settings of adaptation and fine-tuning:

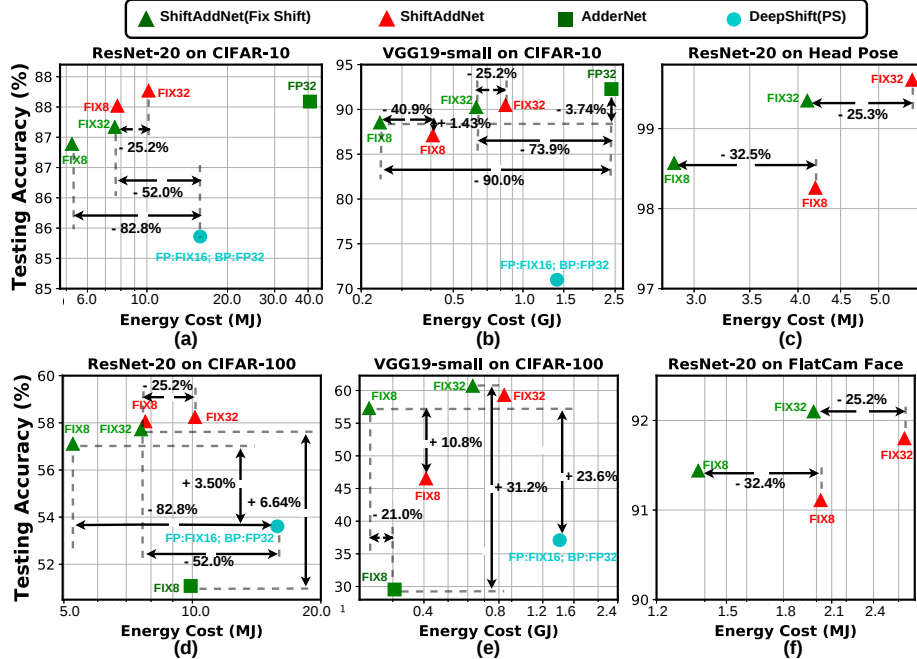

Figure 4: Testing accuracy vs. energy cost of ShiftAddNet **with fixed shift layers** over AdderNet [5] (add only), DeepShift [4] (shift only), and multiplication-based ConvNet [45], using the ResNet-20 and VGG19-small models on the CIFAR-10/100 and two IoT datasets.

Table 2: Adaptation and fine-tuning results comparisons using ResNet-20 trained on CIFAR-10.

| Setting | Methods | Accuracy (%) | | Energy Costs (MJ) |
| | | Adaptation | Finetuning | |
|---|---|---|---|---|
| ResNet20 on CIFAR10 | DeepShift | 58.41 | 51.31 | **9.88** |
| | AdderNet | 79.79 | 84.23 | 25.41 |
| | ShiftAddNet | 81.50 | 84.61 | 16.82 |
| | ShiftAddNet (Fixed) | **85.10** | **84.88** | 11.04 |

- **Adaptation.** We split CIFAR-10 into two non-overlapping subsets. We first pre-train the model on one subset and then retrain it on the other set to see how accurately and efficiently they can adapt to the new task. The same splitting is applied to the test set.

- **Fine-tuning.** Similarly, we randomly split CIFAR-10 into two non-overlapping subsets, the difference is that each subset contains all classes. After pre-training on the first subset, we fine-tune the model on the other, expecting to see a continuous growth in performance.

Tab. 2 compares the testing accuracies and training energy costs of ShiftAddNet and the baselines. We can see that ShiftAddNet always achieves a better accuracy over the two SOTA multiplication-less networks. First, compared to AdderNet, ShiftAddNet boosts the accuracy by 5.31% and 0.65%, while reducing the energy cost by 56.6% on the adaptation and fine-tuning scenarios, respectively; Second, compared to DeepShift, ShiftAddNet notably improves the accuracy by 26.69% and 33.57% on the adaptation and fine-tuning scenarios, respectively, with a marginally increased energy (10.5%).

## 4.4 Ablation studies of ShiftAddNet

We next study ShiftAddNet's shift and add layers for better understanding this new network.

### 4.4.1 ShiftAddNet: fixing the shift layers or not

**ShiftAddNet with fixed shift layers.** In this set of experiments, we study ShiftAddNet with the shift layers fixed or learnable. As shown in Fig. 4, we can see that (1) Overall, ShiftAddNet with fixed shift layers can achieve up to 90.0% and 82.8% energy savings than AdderNet (with floating-point or fixed-point precisions) and DeepShift, while leading to comparable or better accuracies (-3.74% ~ +31.2% and 3.5% ~ 23.6%), respectively; and (2) interestingly, **ShiftAddNet with fixed shift layers also surpasses the generic ShiftAddNet** from two aspects: First, it always demands less energy

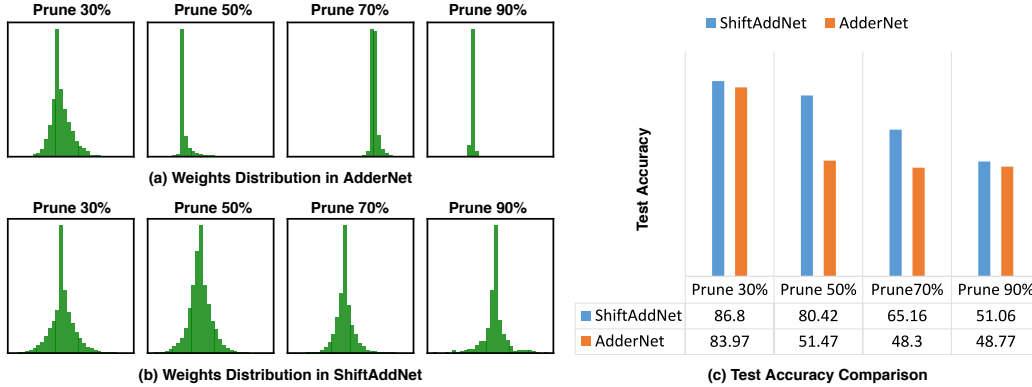

Figure 6: *Left:* Histograms of the weights in the 11-th add layer of ResNet20 trained on CIFAR-10. *Right:* Comparing accuracies of ShiftAddNet with AdderNet under different pruning ratios.

costs (25.2% $\sim$ 40.9%) to achieve a comparable or even better accuracy; and second, it can even achieve a better accuracy and better robustness to quantizaiton (up to 10.8% improvement for 8-bit fixed-point training) than the generic ShiftAddNet with learnable shift layers, when evaluated with VGG19-small on CIFAR-100.

**ShiftAddNet with its fixed shift layers pruned.**
As it has become a common practice to prune multiplication-based DNNs before deploying into resource-constrained devices, we are thus curious whether this can be extended to our ShiftAddNet. To do so, we randomly prune the shift layers by $30\%, 50\%, 70\%$ and $90\%$, and compare the testing accuracy versus the training epochs for both the pruned ShiftAddNets and its corresponding Adder-Net. Fig. 5 shows that ShiftAddNet maintains its fast convergence benefit even when the shift layers are largely pruned (e.g., up to 70%).

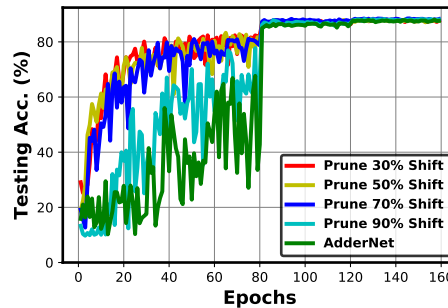

Figure 5: Testing accuracy vs. training epochs for the AdderNet [5] and pruned ShiftAddNets on ResNet-20 with CIFAR-10.

### 4.4.2 ShiftAddNet: sparsify the add layers or not

Sparsifying the add layers allows us to further reduce the used parameters and save training costs. Similar to quantization, we observe that even slightly pruning AdderNet incurs an accuracy drop. As shown in Fig. 6, we visualize the distribution of weights in the 11-th add layer when using a ResNet-20 as backbone under different pruning ratios. Note that only non-zero weights are shown in the histogram for better visualization. We can see that networks with only adder layers, i.e., AdderNet, fail to provide a wide dynamic range for the weights (collapse to narrow distribution ranges) at high pruning ratios, while ShiftAddNet can preserve a consistently wide dynamic ranges of weights. That explains the improved robustness of ShiftAddNet to sparsification. The test accuracy comparisons in Fig. 6 (c) demonstrate that when pruning 50% of the parameters in the add layers, ShiftAddNet can still achieve 80.42% test accuracy while the accuracy of AdderNets collapses to 51.47%.

## 5 Conclusion

We propose a multiplication-free ShiftAddNet for efficient DNN training and inference inspired by the well-known shift and add hardware expertise, and show that ShiftAddNet achieves improved expressiveness and parameter efficiency, solving the drawbacks of networks with merely shift and add operations. Moreover, ShiftAddNet enables more flexible control of different levels of granularity in the network training than ConvNet. Interestingly, we find that fixing ShiftAddNet's shift layers even leads to a comparable or even better accuracy for over-parameterized networks on our considered IoT applications. Extensive experiments and ablation studies demonstrate the superior energy efficiency, convergence, and robustness of ShiftAddNet over its add or shift only counterparts. We believe many promising problems are still open to be discussed for our proposed new network, an immediate future work is to explore the theoretical ground of such a fixed regularization.

## Broader impact

**Efficient DNN training goal.** Recent DNN breakthroughs rely on massive data and computational power. Also, the modern DNN training requires massive yet inefficient multiplications in the convolution, making DNN training very challenging and limiting the practical applications on resource-constrained mobile devices. First, training DNNs causes prohibitive computational costs. For example, training a medium scale DNN, ResNet-50, requires ten to the power of eighteen floating-point operations or FLOPs [51]. Second, DNN training has raised pressing environmental concerns. For instance, the carbon emission of training one DNN can be as high as one American cars' life-long emission [52, 50]. Therefore, efficient DNN training has become a very important research problem.

**Generic hardware-inspired algorithm.** To achieve the efficient training goal, this paper takes one further step along the direction of multiplication-less deep networks. by drawing a very fundamental idea in the hardware-design practice, computer processors, and even digital signal processing. It has been known for long that multiplications can be performed with additions and logical bit-shifts [6], whose hardware implementation is very simple and much faster [8], without compromising the result quality or precision. The above clever "shortcut" saves arithmetic operations, and can readily be applied to accelerating the hardware implementation of any machine learning algorithm involving multiplication (either scalar, vector or matrix). But our curiosity is well beyond this: *we are supposed to learn from this hardware-level "shortcut", for designing efficient learning algorithms.*

**Societal consequences.** Success of this project enables both efficient online training and inference of state-of-the-art DNNs in pervasive resource-constrained platforms and applications. As machine learning powered edge devices have penetrated all walks of life, the project is expected to generate tremendous impacts on societies and economies. Progress on this paper will enable ubiquitous DNN-powered intelligent functions in edge devices, across numerous camera-based Internet-of-Things (IoT) applications such as traffic monitoring, self-driving and smart cars, personal digital assistants, surveillance and security, and augmented reality. We believe the hardware-inspired ShiftAddNet is a significant efficient network training methods, which would make an impact to the society.

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
