[Supplementary Material]

# Supplementary Material of ShiftAddNet

**Haoran You**[†], **Xiaohan Chen**[‡], **Yongan Zhang**[†], **Chaojian Li**[†], **Sicheng Li**[◇], **Zihao Liu**[◇],
**Zhangyang Wang**[‡], and **Yingyan Lin**[†]

[†]Department of Electrical and Computer Engineering, Rice University
[‡]Department of Electrical and Computer Engineering, The University of Texas at Austin
[◇]Alibaba DAMO Academy

## 1   Visualize the divergence of different classes

To further analyze the improved robustness (for quantizing) of combining two weak players: shift and add. we compare the discriminative power of AdderNet and ShiftAddNet. Specifically, we visualize the class divergences using the $t$-distributed stochastic neighbor embedding (t-SNE) algorithm [4], which is well-suited for embedding high-dimensional data for visualization in a low-dimensional space of two or three dimensions (2/3-D). After reducing the dimensions of learned features to 2/3-D, we are then able to analyze the discrimination among different classes, which further allows us to compare the effectiveness of different networks. As shown in Fig. 1, the 2/3-D visualization show that **the proposed ShiftAddNet discriminate different classes better** (i.e., the boundary among different classes can be easier to identify as compared to AdderNet [2]) for both the floating point (FP32) and fixed point (FIX8) scenarios.

Figure 1: T-SNE visualization of the class divergences in AdderNet [2], and the proposed ShiftAddNet, using ResNet-20 on CIFAR-10 as an example.

## 2   Evaluation on two more IoT datasets.

**Accuracy and training costs tradeoffs.** We evaluate DCNN [3] on the popular MHEALTH [1] and USCHAD [5] IoT benchmarks. As shown in Fig. 2 (a) and (b), ShiftAddNet again consistently outperforms the baselines under all settings in terms of efficiency-accuracy trade-offs: (1) **over AdderNet**: ShiftAddNet reduces 32.8% ~ 90.6% energy costs while resulting comparable accuracies

Figure 2: Testing accuracy vs. energy cost comparisons on IoT datasets.

(-0.65% ~ 9.87%); and (2) **over DeepShift**: ShiftAddNet achieves 7.85% ~ 30.7% higher accuracies while requiring 44.1% ~ 74.7% less energy costs.

**Inference costs:** As shown in Fig. 2 (a), when training DCNN on IoT dataset, ShiftAddNet with fixed shift layers (FIX32) costs 1.7 J, where AdderNet (FIX32) costs 1.9 J and DeepShift (FIX32) costs 2.6 J, respectively, leading to 10.5% / 34.6% savings.