[Reviews · NeurIPS 2020]

Review 1

Summary and Contributions: The authors draw a very interesting inspiration, from energy-efficient hardware implementation, i.e., multiplication can be replaced equivalently with additions and logical bit-shifts. Using this important insight, the authors explicitly parameterized deep network, yielding a new type of deep network called ShiftAddNet that involves only bit-shift and additive weight layers.

Strengths: This seems to be a strong work: high originality, insightful and well-executed. -The paper describes a very novel and refreshing line of ideas. While hardware-aware DNNs (algorithms “adapted” for certain hardware) is a hot-enough area, the uniqueness of this work is to have “re-invented” a deep network, that organically “mirrors” a successful principle from hardware design. This is brand-new, and immediately contributes to more energy-efficient inference/training. Moreover, it could yield broader inspirations on new possibilities of deep network blocks (free of convolution, even no multiplication). -AdderNet [6] at CVPR’20 seems to be a cornerstone of this work. But this submission makes well appreciated progress beyond AdderNet, by introducing two complementary light-weight operations types (bit-shift and add) in one framework, which is for the first time AFAIK. It is backed up by two clear motivations: (1) the resultant representation is more efficient than using add only; (2) the bit-shift and add operations naturally correspond to coarse- and fine-grained input manipulations, leading to some novel training trade-off (see next). -The authors developed both inference and training algorithms for their re-designed network. Especially Intriguingly, the authors show by freezing all bit-shifts and only training add layers, ShiftAddNet can still have competitive accuracy compared to existing convolution or multiplication-free networks, while saving aggressively on energy. That sheds more new insights on DNN weight space’s potentially hierarchical latent structure, related to NN landscape and generalization, etc. -Experiments are in general quite thorough and convincing. Overall, ShiftAddNet can reduce over 80% the hardware-quantified energy cost of DNN training and inference, while offering comparable or better accuracies. Only two relatively small models and a few limited-scale datasets are considered, but that’s understandable as the authors mainly claim for energy efficiency benefit in edge computing. One bonus highlight is that the authors implemented ShiftAddNet on a FPGA board and collect all real on-board measurements, closing their “hardware-inspired” loop. The ablation study in Section 4.4 is also good.

Weaknesses: Currently I am giving a score 8, mainly because the idea, motivation and storyline are exciting. But the draft’s Sections 3 & 4 remain unclear in several ways. My final score will depend on how the authors clarify the main questions below: -Section 3 appears to be too “high level” (it shouldn’t be, for the many new things discussed). For example, I was expecting to see how backpropagation was done for the two new layers, but they were unexplained (not even in the supplementary). Also, it is surprising that “fixing shift” as an important extension towards the authors’ claimed “coarse/fine flexibility” only takes five lines in Section 3. A true gem may be overlooked! -Section 4: it is totally unclear what are the dimensions of shift and add layers? For example, when you compare “ShiftAddNet” with ResNet-20, shall I imagine either shift or add layer to have the same dimension as the conv layer, for each layer? Or else? How about DeepShift/AdderNet? Are they fair-comparable to ShiftAddNets in layer/model sizes? - Section 4: The two IoT datasets (FlatCam Face [26], Head-pose detection [11]) are unpopular, weird choices. The former is relatively recent but not substantially followed yet. The latter was published in 2004 and was no longer used much recently. I feel strange why the authors choose the two uncommon datasets, that makes their benchmarking results a bit hard to sense and evaluate. There should have been better options for IoT benchmarking, such as some wearable health or mobile activity recognition data, or even some sets in UCI.

Correctness: Yes

Clarity: The overall writeup quality is okay and can improve more by revision. Specifically, the abstract and introduction parts are truly impressive, having very mature writing style and laying out a convincing logic. But somehow, Sections 2 and 3 start to be unnecessarily wordy on the motivation stuff, while missing some important details - see weaknesses.

Relation to Prior Work: Yes

Reproducibility: Yes

Additional Feedback:


Review 2

Summary and Contributions: This work improve the entering efficiency for deep learning applications, by co-designing DL models to better suit underlying HW capabilities. The key thesis being to avoid the costly multiplications, and replace these operations with shifts (power of two multiplications) and addition operations. Instead of doing this post-facto on an existing model, the approach adopted here is to construct the model bottoms-up with these constrains. Which would make the model energy efficient by design. Proposing a new model architecture - ShiftAddNet constructed with shift and add layers, with a comprehensive methodology for capturing the HW costs for these individual layers as well as the full network. Furthermore, this work also attempts to provides an algorithmic explanation/basis for this alternate model construction. There has been a few efforts along these lines in the recent past. However with this work, build on these efforts to better incorporate the HW costs, consider both training and inference - hence better overall performance and key differentiator being that the proposed idea is prototyped on a FPGA and measurement indicate close correlation with expected results/gains.

Strengths: - The paper is well written, and the analysis presented is quite comprehensive considering both the HW and algorithmic aspects - The authors present a detailed analysis (with ablation studies) and rationale, with the analytical basis. Reasoning about both accuracy and HW efficiency. - And where applicable they do the due-dillegence of comparing with different variants (section 3.4) of the proposed formulation. - Attempt to incorporate recent complimentary works in this area

Weaknesses: - The results/analysis albeit being detailed and comprehensive, only two relatively old and small models are evaluated. - Some of the comparison with other related works, is not completely apple-to-apples, for instance comparing fixed point representation for training, while comparing against AdderNet and DeepShift which use at least half floating points for training. Its understandable the fixed point representation has benefits, but it would probably have been more relevant to compare against similar such fixed point training, for instance other works such (not limited to) - https://arxiv.org/abs/1802.00930, https://arxiv.org/abs/1802.04680, https://arxiv.org/abs/1909.02384 - While the FPGA implementation and results with that are quite impressive and is very valuable as a prototype for the proposed method. However, for such HW solution it would help if the authors extend to do a wider comparison. For instance, with a dedicated ASIC-based implementation the quantum of benefits (table.2) would be considerably reduced. Since fp multiplication could still be cheaper since most optimizations would require non-trivial changes to the datapath, which would take away for the benefits of the faster computations

Correctness: The methodology employed in this paper are largely accurate and correct, along with the claims being made

Clarity: The paper is well written, and easy to follow.

Relation to Prior Work: Given that his work is proposing Hw-driven model/algorithm design, it would be relevant to extend the comparison with wider set of literature, and ensuring a more relevant comparison (more details in the above section).

Reproducibility: Yes

Additional Feedback: One slightly tangential note, it would also help while considering FPGA based solutions to look at the broader scope of applicability of such a solution. Given that the practical application of FPGAs esp. for training at a practical scale is not as popular, primarily given that fact that it is limited by the other platform components such as NW, IO… which are very important for training.


Review 3

Summary and Contributions: The authors explore a combination of both shift layers (i.e. power-of-two weights) and adder layers (i.e. like AdderNet, replace convolutions with additions and take the L1-norm between filters and input feature). They replace all layers with two new layers: bit-shift and add. ShiftAddNet is evaluated against AdderNet and DeepShift on a FPGA. The costs and performance of on-device learning is evaluated.

Strengths: Hardware friendly implementations of neural networks are obviously of interest. This work suggests meaningful gains may be possible by combining both shift and adder layers. Results for on-device training scenarios are of interest too.

Weaknesses: The paper is a little difficult to follow in places. I have some questions about the experimental setup, what has been compared and the results.

Correctness: How is the FPGA's energy consumption measured? Is the hardware cost (area) approximately the same for all implementations? and clock frequency/throughput etc.? I assume Figure 2. reports training energy costs vs. accuracy. Do you report inference costs? Can the training energy costs be brocken down into the major components? For Figure 2 and Figure 4 what are the different networks implementations being compared? * FP32 ConvNet, but no fixed point versions? * AdderNet FP32, FIX32, FIX8, but no FIX16? FP16? * DeepShift - FIX16 only? * ShiftAddNet - FIX8, FIX32, FP32 - but no FIX16, FP16? What does it mean when one of these data points is missing from Figure 2 or Figure 4? What does the comment "under a comparable energy cost" (line 202) mean precisely? Might better results be possible if we choose the datatype and quantization level on a per layer basis? e.g. a hybrid scheme? Is Figure 4 referenced/described in the text? What would these results look like for larger models?

Clarity: I found the early text a little repetitive. The description of the experiments performed could have been clearer.

Relation to Prior Work: The related work section is quite short. What about other approaches to low-cost on-device training? e.g. "E2-Train: Training State-of-the-art CNNs with Over 80% Less Energy", NeurIPS'19

Reproducibility: No

Additional Feedback:

[Author Response · NeurIPS 2020]

**[R1] Methods clarification.** Sorry for not having made it clear enough. ShiftAddNet adopt SOTA bitwise-shift-based
and add-based networks' design to do backpropagation (Line 176). We appreciate your suggestions and will include
detailed formulation/explanation for both the backpropagation and the "fixing shift" extension in the finial revision.

**[R1] Dimensions of shift/add layers.** The shift layer shares the same dimensions with the original ConvNet, followed
by the add layer which adapts kernel sizes / input channels to match the reduced feature maps. Although in this way,
ShiftAddNet has slightly more weights than ConvNet/AdderNet ($\sim$1.3MB for ShiftAddNet with ResNet20 (FP32)
vs. 1.03 MB in the corresponding ConvNet/AdderNet, which can be further quantized to $0.4$ MB without hurting the
accuracy), it takes less energy costs to achieve similar accuracies (Sec. 4.2). Since data movement is the cost bottleneck
in network training/inference as **R2** mentioned (also see Tab. 1), ShiftAddNet makes an important positive step.

**[R1, R2] Evaluation on two popular IoT datasets.** Fol-
lowing your kind suggestion, we evaluate DCNN [Jiang et al.
MM'15] on the popular MHEALTH [Banos et al. IWAAL'14]
and USCHAD [Zhang et al. UbiComp'12] IoT benchmarks.
As shown in Fig. 1 (a) and (b), ShiftAddNet again consis-
tently outperforms the baselines under all settings in terms
of accuracy-cost tradeoffs: (1) **over AdderNet**: ShiftAddNet

Figure 1: Accuracy vs. energy cost comparison.

reduces 32.8% $\sim$ 90.6% energy costs while resulting comparable accuracies (-0.65% $\sim$ 9.87%); and (2) **over DeepShift**:
ShiftAddNet achieves 7.85% $\sim$ 30.7% higher accuracies while requiring 44.1% $\sim$ 74.7% less energy costs.

**[R2,R3] Larger models.** As **R1** kindly mentioned, we mainly claim for energy efficiency benefit in edge computing
using ResNet/VGG on CIFAR/IoT, which are popular benchmarks widely used in latest efficient CNN training papers.
Furthermore, as requested we try larger models and datasets (ResNet-18/34 on ImageNet): ShiftAddNet (63.1% /
68.3%) using ResNet-18/34 architectures improves up to 3.4% top-1 accuracy than AdderNet (59.7% / 64.8%) and
DeepShift (63.2% / 68.1%), with slightly higher energy overheads. Due to limited time, we train both AdderNet and
ShiftAddNet with less epochs & larger batch sizes (fair comparison), after privately consulting AdderNet authors.

**[R2] Completely apple-to-apple comparison.** Thank you for pointing out this and providing references. We follow
your advice to apply quantization training for both AdderNet and DeepShift, and compare ShiftAddNet with them in an
apple-to-apple manner: Evaluated on VGG-19 with CIFAR-10 (see Fig. 1 (c)), ShiftAddNet consistently (1) improves
accuracuies by 11.6%, 10.6%, 37.1% as compared to AdderNet in terms of FIX-32/16/8 formats, with comparable
energy costs (-25.2% $\sim$ 15.7%); and (2) improves accuracies by 26.8%, 26.2%, 24.2% as compared to DeepShift (PS)
in terms of FIX-32/16/8 formats, with comparable or slighly higher energy overheads. Such advantages (robustness for
quantization) can also generalize to other model and dataset pairs, and we will report all of them in the final revision.

**[R2] Training with FPGA.** FPGA is gaining increasing population for both research (e.g., FPGA-based training frame-
work [W. Zhao. ASAP'16]) and next-generation industrial AI (e.g., Intel FPGA acceleration [E. Chung. MICRO'18]).

**[R2] Wider comparisons using ASIC&FPGA.** We follow your sugges-
tion to supply a comprehensive comparison using both ASIC and FPGA,
and analyze the energy savings from both the operation and model per-
spectives (see Tab. 1). Addition and bit-wise shift help to save 1.1$\times$ $\sim$
7.6$\times$ and 3.8$\times$ $\sim$ 9.9$\times$ energy costs over multiplication based ConvNet,
respectively, where the FPGA energy is measured on board and ASIC
energy costs are measured using a SOTA predictor [Xu et al. FPGA'20].

Table 1:Wider comparisons using ASIC&FPGA.

| Inference Type | Format | | ASIC (45nm) | | FPGA | |
|---|---|---|---|---|---|---|
| | Operation | Format | Energy (pJ) | Improv. | Energy (pJ) | Improv. |
| Operation energy | Mult. | FP32 | 3.7 | - | 18.8 | - |
| | | FIX32 | 3.1 | - | 19.6 | - |
| | | FIX8 | 0.2 | - | 0.2 | - |
| | Add | FP32 | 0.9 | 4.1x | 0.4 | 47x |
| | | FIX32 | 0.1 | 31x | 0.1 | 196x |
| | | FIX8 | 0.03 | 6.7x | 0.1 | 2x |
| | Shift | FIX32 | 0.13 | 24x | 0.1 | 196x |
| | | FIX8 | 0.024 | 8.3x | 0.025 | 8x |
| | Operation | Format | Energy (MJ) | Improv. | Energy (GJ) | Improv. |
| Model energy (VGG-19 small) | Mult. | FP32 | 8.08 | - | 3.6 | - |
| | | FIX32 | 7.35 | - | 2.27 | - |
| | | FIX8 | 1.54 | - | 2.27 | - |
| | Add | FP32 | 6.17 | 1.3x | 2.4 | 1.5x |
| | | FIX32 | 5.76 | 1.3x | 1 | 2.3x |
| | | FIX8 | 1.44 | 1.1x | 0.3 | 7.6x |
| | Shift | FIX32 | 0.87 | 8.5x | 0.6 | 3.8x |
| | | FIX8 | 0.21 | 7.3x | 0.23 | 9.9x |

**[R3] ❶ FPGA measurement:** We measure the dynamic power (by
power meter) and latency for one iteration, and then scale the energy
costs to the whole training process; **❷ Hardware area and throughput:**
We by default ensure the hardware cost (area) approximately the same for
all: a default frequency of 100MHz and a throughput of 13FPS / 20FPS
for FIX-32/8 using ResNet-20 on CIFAR; **❸ Inference costs:** E.g., when training DCNN on IoT dataset (see Fig. 1
(a)), ShiftAddNet (FIX-32; fix shift) costs 1.7 J, where AdderNet (FIX-32) costs 1.9 J and DeepShift (FIX-32) costs
2.6 J), respectively, leading to 10.5% / 34.6% savings; **❹ FPGA energy breakdown:** E.g., Clocks: 7%, Signals: 6%,
Logic: 5%, BRAM: 10%, DSP: 1%, PS7: 71%, for ShiftAddNet (FIX-8) with ResNet-20 on CIFAR; **❺ Complete
comparisons:** We supply the additional cases as you suggested, e.g., when training VGG-19 on CIFAR-10 (see Fig. 1
(c)), ShiftAddNet reduces -25.3% $\sim$ 83.1% energy costs over AdderNet, while offering comparable accuracies (-5.17%
$\sim$ 37.12%), and meanwhile achieves 16.1% $\sim$ 24.2% higher accuracies, while reducing -43.6% $\sim$ 70.9% energy costs
over DeepShift; **❻ Comparable energy costs (line 202):** It precisely means ConvNet costs $\pm$30% more than AdderNet
(FP32); **❼ Mixed quantization:** We follow [Elthakeb et al. MICRO'20] to try mixed precision training methods for
ShiftAddNet (Acc.: 88.5%) vs. 88.2% with FIX-32, energy: 28.8% savings over FIX-32; **❽ Fig. 4 reference:** Sorry
for the missing reference, we will add it in Sec. 4.4.1. We appreciate all of these questions and promise to supply
experiments on all the above settings and over E$^2$Train in the final revision.

[Meta-Review · NeurIPS 2020]

This paper was discussed by reviewers after reading the authors’ feedback. Reviewers agree that the feedback addressed some of the concerns. Overall reviewers feel positive about the paper. Please improve the paper according to reviewers’ feedback in the final camera ready version.